# Antioxidant Properties and Structure-Antioxidant Activity Relationship of *Allium* Species Leaves

**DOI:** 10.3390/molecules26237175

**Published:** 2021-11-26

**Authors:** Dikdik Kurnia, Dwipa Ajiati, Leny Heliawati, Dadan Sumiarsa

**Affiliations:** 1Department of Chemistry, Faculty of Mathematics and Natural Science, Universitas Padjadjaran, Sumedang 45363, Indonesia; dwipa20001@mail.unpad.ac.id (D.A.); dadan.sumiarsa@unpad.ac.id (D.S.); 2Study Program of Chemistry, Faculty of Mathematics and Natural Science, Universitas Pakuan, Bogor 16143, Indonesia; leny_heliawati@unpak.ac.id

**Keywords:** *Allium* species leaves, antioxidant properties, SAR

## Abstract

*Allium* is a genus that is widely consumed and used as traditional medicine in several countries. This genus has two major species, namely cultivated species and wild species. Cultivated species consist of *A. cepa* L., *A. sativum* L., *A. fistulosum* L. and *A. schoenoprasum* L. and wild species consist of *A. ursinum* L., *A. flavum* L., *A. scorodoprasum* L., *A. vineale* L. and *A. atroviolaceum* Boiss. Several studies report that the *Allium* species contain secondary metabolites such as polyphenols, flavonoids and tannins and have bioactivity such as antioxidants, antibacterial, antifungal, anti-inflammatory, pancreatic α-amylase, glucoamylase enzyme inhibitors and antiplatelets. This review summarizes some information regarding the types of *Allium* species (ethnobotany and ethnopharmacology), the content of compounds of *Allium* species leaves with various isolation methods, bioactivities, antioxidant properties and the structure-antioxidant activity relationship (SAR) of *Allium* compounds.

## 1. Introduction

Antioxidants play important roles in health. They are also used to reduce disease risk and have the ability to protect the body against oxidative damages, which cause several diseases (diabetes, cancer, and neurodegenerative disorders, etc.). Antioxidants can control oxidative processes, leading to food quality descent caused by reactive oxygen species (ROS) and free radical reactions in the body [1,2].

ROS and free radicals are the main causes of oxidative stress, which can trigger several degenerative diseases such as cancer, coronary heart disease and vascular disease [3]. Based on the dangers posed by ROS and free radical reactions, it is necessary to have natural antioxidants that can prevent oxidative stress. Several studies reported that the leaves of the *Allium* species have good antioxidant activity [4,5,6].

*Allium* is a genus of the *Liliaceae* family easily found in Asia, Europe and America [7]. This genus has more than 700 species such as *A. cepa* L., *A. sativum* L., *A. fistulosum* L., *A. schoenoprasum* L. *A. ursinum* L., *A. flavum* L., *A. scorodoprasum* L., *A. vineale* L., *A. atroviolaceum* Boiss., *A. psekemense* B. Fedtsch., *A. kurtzianum*, *A. chinense* and *A. rubellum* and many other species from other countries [8,9]. In the last 10 years, several studies have reported that *Allium* contains several secondary metabolites in the bulbs, flowers and leaves [6,10,11,12,13]. Secondary metabolites are rich in health benefits because they have several bioactivities such as antioxidant, anticancer, antibacterial, antifungal, anti-inflammatory and anti-platelet [5,14,15,16,17,18,19,20,21,22,23,24,25,26]. Based on this, this review focused on the antioxidant activity found in the leaves of nine *Allium* species that have been extensively studied [27]. This review will also discuss the relationship between antioxidant activity and the structure of several compounds contained in *Allium* leaf extract.

## 2. Ethnobotany and Ethnopharmacology of *Allium* Species

*Allium cepa* L., onions, is the most cultivated plant widely used as a spice because of its distinctive taste and aroma [28]. The plant has fibrous roots and consists of 3–8 leaves. The base of the leaves is fleshy and forms bulbs that are round or elongated according to the variety [29]. *A. cepa* has several types of varieties, namely yellow onions, white onions, red onions and green onions [30]. This plant is easy to grow in areas with fine, rock-free and well-draining soil types. In addition, this species is known to have a hypocholesterolemic effect and can prevent heart disease [31].

*Allium sativum* L., garlic, is an annual plant that has been cultivated about 5000 years ago in the Middle East, is easy to grow in cold areas and has bulbs consisting of several cloves. There are 4–40 cloves in each stem. The bulb can grow to a size of 25–70 cm with long, flat and folded leaves [30,31]. The flowers are greenish-white and about 3 mm in length [32,33]. Several studies reported that this plant contains compounds that have low toxicity, for example, sulfur compounds such as allicin and fructose polymers [34]. As well as *A. cepa* L., the plant is also reported to have heart disease-preventing effect [31].

*Allium fistulosum* L., known as Japanese bunching onions, welsh onions or spring onions is one of *Allium* species known for originating from Romania which has similarities to scallions and smells and tastes similar to *A. cepa* L. Unlike other species, this plant does not form bulbs and has hollow leaves [35]. In Japan and China, this plant is used as a vegetable or as a traditional medicine to improve the function of internal organs and metabolism and treat several diseases such as headaches, diarrhea, stomach pains and colds [32,36].

*Allium schoenoprasum* L., chives, is one of the important spice plants in Central Europe and is widely cultivated in several countries such as Austria, France, Germany and the Netherlands. Morphologically, this plant can adapt easily to dry and sunny habitats [37]. This species has a thin bulb, shaped like a cone with a length of 2–3 cm and 1 cm in length. The leaves are hollow and can be consumed because they have a mild onion taste. Another characteristic of this plant is that it has a soft, tube-shaped hollow stem with a diameter of 23 mm and purple flowers. In some countries, this species is used as traditional medicine. In Indonesia, *A. schoenoprasum* is used to treat hypertension. Meanwhile, in East Asia, this is used to relieve flu and lung congestion [38]. 

*Allium ursinum* L., known as bear garlic, ramsons or wild garlic, is mostly found in forest areas close to rivers. There are also those who call it forest garlic [39]. This plant can grow to a height of 50 cm with white flowers and bulbs of a size of less than 6 cm [15]. In the middle of the century, the plant is used to treat dyspepsia, cardiovascular disease, cancer, obesity and diabetes [40].

*Allium flavum* L., known as small yellow onion, is a species native to Southern, Central and Eastern Europe and Western Asia. This plant has leaves and bulbs that can be consumed and commonly used as a spice in several dishes such as salads, soups and stews [10]. This species has a height of about 10–60 cm with round elongated bulbs at the top. Other characteristics include having lanceolate-shaped leaves with stripes up to 2 mm long and small yellow flowers. This plant commonly grows in rocky meadows in the Mediterranean region and is used as traditional medicine because of its strong hepatoprotective, immunostimulant and antihypertensive properties [41].

*Allium scorodoprasum* L., wild leek, is an annual plant widely grown in the North and East Anatolia region which is commonly used as a mixture in the manufacture of cheese, yogurt and bread [42]. This species is 25–90 cm tall with bulbs 1–2 cm in diameter. In addition, this plant has leaves of 2–5 strands, each of a diameter of 2–8 mm, and flowers are dark red and purple. Morphologically, this species is adaptable in calcareous areas, grasslands, rural areas and clay slopes. Pharmacologically, this plant is used as an antiseptic, wound healing, treating hypotension and diuretic and can prevent aging, cardiovascular disease and liver disease [43].

*Allium vineale* L. known as crow garlic is a plant that is considered as a weed and originates from Eurasia. The 30–100 cm tall plant has membranous bulbs and linear leaves. The first leaf, with a length of 3–5 cm, will become the midrib and the next leaf will cover the previous leaf blade. This species is commonly used as traditional medicine in curing several diseases such as pneumonia, ulcers, bronchitis, digestive disorders, and others [44]. *A. vineale* bulb can be used as a substitute for *A. sativum* in cooking, while the leaves are used as a salad [45].

*Allium atroviolaceum* Boiss. is an annual herb originating from Zagros, a region in Iran [46]. This species is used as a vegetable and also a source of vitamins [47].

## 3. Isolation Methods and Compound Content of *Allium* Species Leaves

### 3.1. Allium cepa

Several studies report the isolation of *A. cepa* with many methods. Samples were extracted using the solvent extraction/conventional maceration method with 70% ethanol solvent in a ratio 5:1 to the sample [7]. In another study conducted by Amabye et al., the ethanol extract was characterized by HPLC to determine the amount of phenolics [40]. 

From some different isolation methods, it was also reported that *A. cepa* leaves contain anthocyanin pigments [48], phenolics (catechin, cinnamic acid, ferulic acid, *p*-coumaric acid and sinapic acid) [40], tannins, saponins [49], glycosides/aglycones, quercetin 4-*O*-β-glycopyranoside, quercetin 3,4-*O*-β-diglycopyranoside, quercetin 3,7,4-*O*-β-triglycopyranoside [7], carotenoids and glutathione [27,50] (Figure 1).

### 3.2. Allium sativum

Studies concerning *A. sativum* also use several isolation methods. Sample residue was extracted using petroleum ether, chloroform and ethyl acetate. Then, the chloroform extract was analyzed using silica gel column chromatography (60–120 mesh) with mobile phase CHCl_3_-MeOH (90:10; 80:20; 70:30; 50:50) and guided by TLC with mobile phase CHCl_3_-MeOH (90:10) and iodine vapor was used as a detecting agent. In the CHCl_3_-MeOH fraction (70:30), a single stain of oleanolic acid compound was obtained [49]. Isolation was carried out to obtain methanol, chloroform and aqueous extracts using the conventional maceration method [51]. In addition, five new compounds were obtained in the form of cyclic organosulfur compounds. The five compounds were successfully isolated from the acetone extract, which was partitioned with the EtOAc-H_2_O mobile phase (1:1) and separated by normal phase silica gel column chromatography with CHCl_3_-MeOH mobile phase (1:0; 100:1; 50:1; 30:1; 10:1; 0:1) to produce nine fractions and followed by reverse-phase silica gel column chromatography with MeOH-H_2_O (2:8; 4:6; 6:4; 8:2; 1:0) as mobile phase until thirteen fractions were obtained. Subsequently, the thirteen fractions were purified by HPLC to obtain five pure compounds [52].

Several studies report that *A. sativum* leaves contain a triterpene compound, namely oleanolic acid [53]. The research conducted by Wilson et al. reported that the leaf extract of *A. sativum* contained terpenoids in the methanol extract, glycosides in the chloroform extract and saponins, terpenoids and glycosides in the aqueous extract [51]. In addition, phytochemical tests on aqueous extracts showed the presence of flavonoids, alkaloids, steroids, triterpenes [54], *S*-allyl-L-cysteine, glutamyl-*S*-allyl-L-cysteine, *S*-allyl-L-cysteine sulfoxide (allin) [55,56], foliogarlic disulfane A_1_, foliogarlic disulfane A_2_, foliogarlic disulfane A_3_, foliogarlic trisulfane A_1_, foliogarlic trisulfane A_2_ [52], quercetin 3-*O*-β-D-glucopyranoside (isoquercitrin), quercetin 3-*O*-β-D-xylopyranoside (reynoutrin), kaempferol 3-*O*-β-D-glucopyranoside (astragalin) and isorhamnetin 3-*O*-β-D-glucopyranoside [57] (Figure 2).

### 3.3. Allium fistulosum

Based on research conducted by Fukaya et al., seven compounds were isolated using a simple maceration extraction method with acetone as a solvent. Furthermore, the extract was partitioned with EtOAc-H_2_O (1:1) to obtain the EtOAc fraction then separated using normal phase silica gel column chromatography [CHCl_3_-MeOH (1:0; 200:1; 100:1; 50:1; 0:1)]; [hexane-acetone (1:0; 6:1; 4:1; 3:1; 2:1; 0:1)] and reverse phase [MeOH-H_2_O (6:4; 7:3; 8:2; 9:1; 1:0)]. The last step is purification using HPLC [58]. There is no difference from other methods in the isolation of apigenin compounds carried out by Immaculate V. et al. using the TLC method and purified by HPLC. Aqueous extracts were analyzed by GC-MS [59]. The acetone extract was partitioned with EtOAc-H_2_O as a solvent and separated by silica gel column chromatography with CHCl_3_-MeOH (100:1) as mobile phase to obtain five fractions. Furthermore, fraction 3 was purified again by silica gel column chromatography with *n*-hexane-acetone (6:1) as mobile phase [60].

Several isolation methods of several studies reported that *A. fistulosum* leaf extract contains flavonoids (myricetin, quercetin, rutin, kaempferol, naringenin and hesperetin), polyphenols (benzoic acid, salicylic acid, ferulic acid, caffeine, *p*-coumaric acid, coumarin, vanillic acid, gallic acid and cinnamic acid) [61], kujounin A_3_, kujounin B_1_, kujounin B_2_, kujounin B_3_, allium sulfoxide A_2_, allium sulfoxide A_3_, kujounin A_1_ [58], apigenin (4′,5,7-trihydroxy-flavone) [35], dichloroacetic acid, 1-buten-3-yne, 1-chloro-, (*Z*)-,-pinene, D-limonene, thymol [59], onionin A_1_, onionin A_2_ and onionin A_3_ [60] (Figure 3).

### 3.4. Allium schoenoprasum

In a study conducted by Dominguez et al., *A. schoenoprasum* leaves were extracted using conventional maceration with an ethanol solvent [62]. In some studies, the sample was also extracted using a hydrodistillation method and analyzed by GC-MS [63]. The extract was partitioned with EtOAc-H_2_O (1:1) and isolated by normal phase silica gel column chromatography with CHCl_3_-MeOH (1:0; 200:1; 100:1; 50:1; 0:1) as mobile phase, then, fraction 2 was isolated using reverse-phase silica gel column chromatography with MeOH–H_2_O (6:4; 7:3; 8:2; 9:1; 1:0) as mobile phase and purified by HPLC with H_2_O-MeCN (60:40) as mobile phase [58]. In addition, the Folin–Ciocalteu method was also used to analyze the phytochemical content [64].

*A. schoenoprasum* leaf extract from some studies with different methods reported that these contain *bis*-(2-sulfhydryethyl)-disulfide, 2,4,5-trithiahexane, *tris*-(methylthio)-methane [63], quercetin, kaempferol, myricetin, catechin, rutin, chlorogenic acid, *p*-coumaric acid, ferulic acid, caffeic acid [62,65], folionin A_1_, folionin A_2_, folionin B [66], sitosterol, stigmasterol, campesterol, cholesterol, free fatty acid, monoacylglycerin, diacylglycerin, triacylglycerin, linoleic acid, palmitic acid, spirostanols (deltonin, saponin A), furostanols (deltoside, protodioscin) [67] and I-ketose [68] (Figure 4).

### 3.5. Allium ursinum

Studies concerning *A. ursinum* also use several isolation methods. In a study conducted by Barla et al., a sample was extracted with an aqueous-ethanol solvent for 2 h at 80 °C to obtain a powdered extract [69]. In another study, a sample was extracted using two maceration methods, namely by water infusion and water decoction. Then, the determination of the amount of phenolic content was conducted by the colorimetric assays [70].

From some different isolation methods of several studies, it was reported that *A. ursinum* extract contains *p*-coumaric acid, ferulic acid, kaempferol, kaempferol-3-*O*-glycoside, ursolic acid, quercetin, phenol compound [7,10,71], allicin [21], malondialdehyde (MDA), carotenoids [27,50,68], kaempferol-3,7-di-*O*-β-D-glucopyranoside, kaempferol-(acetylhexoside)-hexoside, acetyl-kaempferol-deoxyhexose [71], propylene sulfide, dimethyldisulfide, dimethylthiophene, (*E*)-methyl-2-propyl disulfide, (*Z*)-methyl-2-propenyl disulfide, dimethyl trisulfide, di-2-propenyl disulfide, 2-vinyl-1,3-dithiane, (*E*)-propenyl propyl disulfide, (*Z*)-propenyl propyl disulfide, methyl-2-propenyl trisulfide, 3,4-dihydro-3-vinyl-1,2-dithiine, 2-vinyl-4H-1,3-dithiine, dimethyl tetrasulfide, (*E*)-di-2-propenyl trisulfide, (*Z*)-di-2-propenyl trisulfide, di-2-propyl trisulfide, di-2-propenyl tetrasulfide [72], inulin, nystose, and I-ketose [68] (Figure 5).

### 3.6. Allium flavum

Rezgui et al. also conducted research concerning *A. flavum*. The study conducted refluxing samples two times for 1 h with MeOH–H_2_O (7:3) [11]. The solvent was evaporated and then VLC (silica gel RP-18, MeOH-H_2_O 50:50) was conducted to produce several fractions to separate using CC (Sephadex LH-20, MeOH). The last step was the MPLC method (RP-18 silica gel, MeOH/H_2_O gradient 40–100%) to obtain pure compounds [11]. In addition, in another study, samples were suspended in 5 mL 1 mol/l K_2_HPO_4_ with pH 7.0 and centrifuged for 10 min to obtain an aliquot of the supernatant, which will be used in the SOD activity test [27].

The results of several studies with different methods reported that *A. flavum* leaves contains three new compounds of spirostane-type glycosides, namely (20*S*,25*R*)-2α-hydroxyspirost-5-en-3β-yl, (20*S*,25*R*)-2α-hydroxyspirost-5-en-3β-yl, (20*S*,25*R*)-spirost-5-en-3β-yl [11], flavonoids and carotenoids [27] (Figure 6).

### 3.7. Allium scorodoprasum

Prior to the antioxidant activity test, Tasci et al. first performed a sample extraction with 80% using an ultrasonic bath [12].

*A. scorodoprasum* leave contains flavonoid and carotenoid compounds [27]. In addition, in research conducted by Tasci et al., the organosulfur compound obtained was allicin [12] (Figure 7).

### 3.8. Allium vineale

Several methods were used to isolate compounds from *A. vineale*. One of the methods used was a sample hydrodistilled first for 4 h and extracted with CH_2_Cl_2_. The filtrate was obtained and then evaporated to remove the solvent to obtain essential oils that have a sharp odor [45]. In other research, a sample *A. vineale* was first boiled using distilled water before being extracted with ethyl acetate and a concentrated organic layer. The extract obtained was then partitioned using silica gel chromatography with hexane, ethyl acetate and methanol as mobile phases to obtain twenty-five fractions [16].

Based on the previous studies using different isolation methods of this species, it contains 2-furaldehyde, (2*E*)-hexenal, (3*Z*)-hexenol, 2,4-dimethylthiophene, allyl methyl disulfide, methyl (*Z*)-1-propenyl disulfide, methyl (*E*)-1-propenyl disulfide, benzaldehyde, dimethyl trisulfide, diallyl disulfide, allyl (*Z*)-1-propenyl disulfide, allyl (*E*)-1-propenyl disulfide, 1-propenyl propyl disulfide, methyl (methylthio)methyl disulfide, allyl methyl trisulfide, 4-methyl-1,2,3-trithiolane, methyl propyl trisulfide, methyl (*Z*)-1-propenyl trisulfide, methyl (*E*)-1-propenyl trisulfide, dimethyl tetrasulfide, allyl (methylthio)methyl disulfide, diallyl trisulfide, allyl (*Z*)-1-propenyl trisulfide, *p*-vinylguaiacol, allyl propyl trisulfide, 5-methyl-1,2,3,4-tetrathiane, methyl (methylthio)methyl trisulfide, allyl methyl tetrasulfide, allyl (methylthio)methyl trisulfide, 4-methyl-1,2,3,5,6-pentathiepane [45], chrysoeriol, chrysoeriol-7-O[2 00-*O*-*E*-feruloyl]-β-D-glucoside and isorhamnetin-3-*O*-β-D-glucoside [16] (Figure 8).

### 3.9. Allium atroviolaceum

Based on the research method conducted by Sebtosheikh et al., samples were cut into small pieces and extracted by hydrodistillation using a Clevenger-a type of apparatus for 4 h. The extract obtained was then evaporated using anhydrous sodium sulfate and stored at 4 °C. Then, the dry extract was analyzed using GC-MS [73,74]. In another research, dry samples were extracted by maceration with H_2_O and 70% ethanol for 48 h. The extract was filtered and evaporated to remove the solvent to obtain water and ethanol extract [46].

Several studies reported that isolation of *A. atroviolaceum* leaves using different methods resulted in several essential oils such as dimethyl trisulfide, ethyl linolenate, phytol [73], norbornene, 3,4-dimethylthiophene, isocitronellene, methyl 1-propenyl disulfide, dimethyl trisulfide furan, *trans*-2-(2-pentenyl)-furan, limonene, diallyl disulfide, 1,3-dithiane, 1,2-dithiolane, methyl (methylthio)methyl disulfide, 4,6-dimethyl-[1,2,3]-trithiane, 5,9-undecadien-2-one, 6,10-dimethyl-(*Z*), 1,1′-thiobis-3-(methylthio)-propane, 4-(2,6,6-trimethyl-1-cyclohexen-1-yl)-3-buten-2-one, 5-methyl-2-phenyl-2-hexenal, 2-methyl-3-oxo-*cis*-cyclohexanebutanal, 1,2,4-cyclopentanetrione, 3-(2-pentenyl)-, di-2-propenyl tetrasulfide, formic acid, 1,2-dithiolane,1.1-dioxide, tetradecanoic acid, 1-(2-ethyl-[1,3]dithian-2-yl)-3-methyl-butan-1-ol and 6,10,14-trimethylpentadecan-2-one [74] (Figure 9).

## 4. *Allium* Species Leave Bioactivity and Test Methods

### 4.1. Antimicrobial

One of the activities found in several species *Allium* is as an antimicrobial. This activity serves as a source of antibiotics against microorganisms such as pathogens and microorganisms that can cause defects in food [75]. Amabye et al. reported that the antimicrobial activity of the ethanol extract of *A. cepa* leaves can inhibit the pathogen *Streptococcus pneumoniae* [40]. *S. pneumoniae* is a pathogen that has an important role in causing invasive diseases such as pneumonia, septicemia, meningitis and some types of eye infections [76,77,78]. An antimicrobial activity assay was carried out using the agar well diffusion with sterile dimethyl sulfoxide (DMSO) as a negative control and gentamicin as a positive control to determine the sensitivity of each bacterial species to be tested. The study showed that there was antimicrobial activity in the ethanol extract of *A. cepa* leaves, which managed to inhibit *S. pneumoniae* with an inhibition zone between 11.87–19.23 mm at 20 mg/mL and the minimum inhibitory concentrations (MIC) value at 10 mg/mL.

In other studies also using the agar cup method, *A. cepa* leaf extract showed good antimicrobial activity in inhibiting the growth of bacteria and fungi such as *Aspergillus* sp., *Botrytis* sp. and *Penicillium* sp., each with an inhibition zone of 817 sq.mm, 817 sq.mm and 377 sq.mm, respectively [79]. Antimicrobial activity was also found in other *Allium* species such as *A. ursinum* [69], *A. sativum* to *Listeria monocytogenes* [78,79] and *A. atroviolaceum* [46]. Krivokapic et al. reported that antimicrobial activity was also present in *A. ursinum* leave. The test was carried out by the microdilution plate method to determine the MIC and minimum microbial concentration (MMC) value. The result indicated that the leaf extract had antimicrobial activity inhibiting the growth of 20 bacteria and fungi [10]. Other studies also state that *A. ursinum* leaf extracts contain organosulfur compounds such as propylene sulfide, (*E*) methyl-2-propenyl disulfide and (*Z*) methyl-2-propenyl disulfide and several other compounds that have antimicrobial activity [72].

#### 4.1.1. Antibacterial

Research conducted by Solomon et al. reported the presence of antibacterial activity in *A. cepa* leaves with different types of extracts, namely ethanol extract, hot extract and cold extract. The three extracts were compared to see the difference in their compound content and the efficiency of their antibacterial activity. The hot extract contains more flavonoids and saponins than the other two extracts. Hot extract also showed the best antibacterial activity among ethanol extract and cold extract in inhibiting the growth of *Escherichia coli*, *Streptococcus* and *Stephylus* [80].

#### 4.1.2. Antifungal

Parvu et al. reported that the ethanol extract of *A. ursinum* leaves has antifungal activity. This study uses the agar dilution assay by determining the MIC value. Although the content of allicin compounds in the leaves is not as much as in the flowers, the antifungal activity in the ethanol extract of *A. ursinum* leaves is able to fight several types of fungi with the MIC value of 120 µL/mL (*A. niger*), 80 µL/mL (*B. cinerea*), 100 µL/mL (*B. paeoniae*), 160 µL/mL (*F. oxysporum* f. sp. *tulipae*),120 µL/mL (*P. gladioli*) and 80 µL/mL (*S. sclerotiorum*) [21].

### 4.2. Antioxidant

Research on antioxidant activity in *Allium* species has been widely reported. Antioxidant activity is found in many *Allium* species with different test methods. Dominguez et al. reported the antioxidant activity of *A. schoenoprasum* based on the total iron reduction’s potential technique [62]. In several studies, the antioxidant activity test of *A. schoenoprasum* and *A. ursinum* used different methods such as DPPH radical-scavenging ability, ferric reducing antioxidant power (FRAP) assay and ABTS radical scavenging assays [17,52,54,55]. The use of some of these methods is intended to compare the results of activities between one method and another and to determine which method is more appropriate to be used in certain species. Parvu et al. did the same, which uses two methods, namely DPPH bleaching method and the trolox equivalent antioxidant capacity (TEAC) assay, where the antioxidant activity of the leaf extract was shown to be higher by the TEAC method than DPPH bleaching method [64].

### 4.3. Anti-Inflammatory

Inflammation is a condition in which catabolism is more dominant or faster than anabolism [81]. Inflammation can also be defined as a reaction to defend the body in eliminating factors that can cause damage and the formation of homeostasis in the body. This causes increased blood flow due to the increased permeability of capillaries and white blood cells to the site of inflammation, resulting in inflammatory symptoms such as redness, swelling and pain [82]. Several previous studies reported that the *Allium* species has several bioactivities, one of which is anti-inflammatory [83,84,85,86,87]. Parvu et al. reported that *A. schoenoprasum* leaf extract with three different concentrations (25, 50 and 100%) had anti-inflammatory activity. The research was conducted by the method of testing in vivo using a turpentine oil-induced inflammation model, while in terms of in vitro, the three extracts were able to inhibit phagocytosis by reducing nitro-oxidative stress [64].

Pan et al. reported that anti-inflammatory activity was also present in *A. sativum* aqueous leaf extract. The extract was previously screened for phytochemicals. The results showed that the extract contained carbohydrates, reducing sugars, lipids, flavonoids, ketones, alkaloids, steroids and triterpenes. The study used two different anti-inflammatory activity testing methods, namely carrageenan-induced paw edema and histamine-induced paw edema. Both methods showed that *A. sativum* aqueous leaf extract was able to lower the paw edema significantly [88]. The same activity is also found in *A. fistulosum* aqueous leaf extract or welsh onions. However, this study only used a carrageenan-induced paw edema method. The results showed that the aqueous extract was able to inhibit the development of paw edema by reducing the activity of the catalase (CAT), superoxide dismutase (SOD) and glutathione peroxidase (GPx) enzymes found in paw edema mice by 43, 74 and 50%, respectively [87].

### 4.4. Antitumor

Tumors are pathological cells that can interfere with cell growth to be abnormal. Tumors, often called neoplasms, are divided into benign tumors and malignant tumors. Malignant tumors are often referred to as cancer [89]. The prevention and cure of tumors can be obtained from natural ingredients that contain compounds that have antitumor activity. A number of *Allium* species are known to have antitumor activity, such as *A. cepa*, *A. sativum*, *A. fistulosum* and *A. schoenoprasum* [90]. Shirshova et al. reported that the aqueous and EtOH-H_2_O extract of *A. schoenoprasum* leaves have antitumor activity. Several compounds such as sitosterol, stigmasterol, campesterol, cholesterol, deltonin, saponin A and mono-, di- and triacylglycerin were isolated and tested for antitumor activity in 40 male BDF mice. EtOH-H_2_O extract of *A. schoenoprasum* leaves was given to mice that have been divided into four groups and given previous treatment and injected with tumor strains *Ehrlich carcinoma* (EC). The results showed that the extract of *A. schoenoprasum* leaves can inhibit tumor growth. The results can be seen by comparing tumor volume and mass between the experimental group and the control group [67].

### 4.5. Antiplatelet

Platelets are small cell fragments that clump in the area of the injured blood vessel [91]. Platelets have an important role in hemostasis, which functions in stopping bleeding [92]. Dysfunctional or abnormal platelets can cause cardiovascular damage, such as myocardial infarctions and strokes. Several studies have developed antiplatelet drugs such as aspirin to overcome the problem of abnormal platelets [93]. Several studies have reported that some *Allium* species have antiplatelet activity. Saplonţai-Pop et al. reported that *A. cepa* bulbs have antiplatelet activity. The activity test used platelet-rich plasma (PRP), which is based on the kinetic curve of decreasing plasma OD [94], while the research conducted by Ko et al. used the platelet aggregation turbidometric assay [95]. Hiyasat et al. reported that *A. ursinum* leaf extract has antiplatelet activity. Testing of antiplatelet activity in vitro was conducted using light transmission aggregometry which has been induced with adenosine diphosphate (ADP), collagen, A23187, epinephrine and arachidonic acid (ARA) [17]. In other species such as *A. fistulosum* and *A. schoenoprasum* was also found the presence of antiplatelet activity using the electrical impedance aggregator method. Although the activities possessed by these two species are not as good as in other species, such as *A. sativum* and *A. ascalonicum* [65]. While *A. atroviolaceum* extract has excellent antiplatelet activity and is able to inhibit platelet aggregation in vitro induced by ARA and ADP with each IC_50_ value of 0.4881 (0.4826–0.4937) and 0.4945 (0.4137–0.5911) mg/mL [74].

### 4.6. Pancreatic α-Amylase and Glucoamylase Enzyme Inhibitor

The reduction of carbohydrate digestion can be carried out by controlling the activity of hydrolysis enzymes, α-amylase and glucoamylase. Controlling this activity can affect postprandial hyperglycemia, which is considered to have a prophylactic healing effect in patients with type 2 diabetes mellitus [96]. The inhibition of hydrolytic enzyme activity is one of the right efforts to suppress carbohydrate digestion and monosaccharide absorption [97]. In human physiology, pancreatic α-amylase is a type of α-amylase which is found in plants, fungi and bacteria [98]. The amount of pancreatic α-amylase synthesized in the rough endoplasmic reticulum is regulated by the amount of food substrates [99], whereas glucoamylase is an enzyme that can be produced from a number of organisms such as *Aspergillus niger* and *Aspergillus awamori*. This enzyme plays a role in producing a certain amount of glucose [100] (Table 1).

Meshram and Khamkar succeeded in isolating oleanolic acid compounds from the chloroform fraction of *A. sativum* leaves using an enzyme activity inhibition test of pancreatic α-amylase and glucoamylase carried out by the Miller method [101], which was modified. The study reported that oleanolic acid showed excellent inhibition of both enzymes. The highest inhibition value occurred at a concentration of 100 µg/mL was 57.50% with IC_50_ 83.56 µg/mL for the glucoamylase enzyme, while it was 62.43% for pancreatic α-amylase with IC_50_ 55.51 µg/mL [53].

**Table 1 molecules-26-07175-t001:** Test Methods for Antioxidant Activity and Other Bioactivities of *Allium* Species Leaves.

Species	Antioxidant Test Methods	Other Bioactivities
*A. sativum*	The phosphomolybdenum reduction assay [102]	Antimicrobial [103,104]; Anti-inflammatory [88]; Inhibitor pancreatic α-amylase and glucoamylase [53]
*A. ursinum*	DPPH radical-scavenging ability [70,71,105]; ferric reducing antioxidant power (FRAP) assay [68]; ABTS radical scavenging assays [23]	Antimicrobial [72,105]; Antifungal [21]; Antiplatelet [17]
*A. schoenoprasum*	DPPH radical-scavenging ability; ferric reducing antioxidant power (FRAP) assay [68]; DPPH bleaching method; the Trolox equivalent antioxidant capacity (TEAC) assay [64]; the total iron reduction’s potential technique [62]; ORAC (oxygen radical absorbance capacity) [65]	Anti-inflammatory [73]; Antitumor [67]; Antiplatelet [65]
*A. fistulosum*	DPPH free radical scavenging assay [61]; ORAC (oxygen radical absorbance capacity) [65]	Anti-inflammatory [87]; Antiplatelet [65]
*A. scorodoprasum*	Ferric reducing/antioxidant power (FRAP); DPPH radical scavenging activity assay [12]	-
*A. vineale*	Ferric thiocyanate method; ferric ions (Fe^3+^) reducing antioxidant power assay (FRAP); DPPH free radical-scavenging activity [16]	-
*A. cepa*	Antioxidant enzyme method [27]	Antimicrobial [40,79]; Antibacterial [80]; Anticardioprotective [49]
*A. flavum*	-	Anticancer [11]
*A. atroviolaceum*	-	Antimicrobial [46]; Antiplatelet [74]

## 5. Antioxidant Properties

Antioxidants are a system to protect our bodies from osulfide, allyl methyl tetrasulfide, allyl (methylthio)methyl trisulfide, 4-mexidative stress caused by free radical and reactive oxygen species (ROS) [48]. Oxidative stress can occur due to the formation of ROS and the detoxification of increased levels of ROS in balance, causing impaired cellular function [106]. Oxidative stress due to ROS can cause several chronic diseases such as cancer, coronary heart disease and osteoporosis. Free radical reactions can attack biomolecules, especially the polyunsaturated fatty acids of cell membranes. ROS which are included as free radicals include superoxide anion (O_2_^•−^), perhydroxyl radicals (HO_2_^•^), hydroxyl radicals (^•^OH) and nitric oxide and other species such as hydrogen peroxide (H_2_O_2_), singlet oxygen (^1^O_2_), hypochloric acid (HOCl) and peroxynitrite (ONOO^−^) [107,108]. The formation of ROS starts from the uptake of O_2_, then activates NADPH oxidase and produces superoxide anion radicals and continues with the conversion of O_2_ that becomes H_2_O_2_ by SOD [109]. Antioxidants break the chain of free radical reactions by donating their own electrons to free radicals without becoming free radicals [106,110].

Based on their activity, antioxidants are classified into two types, which are enzymatic and non-enzymatic. Enzymatic antioxidants are antioxidants that involve several enzymes such as GPx, CAT and SOD in catalyzing free radical and ROS neutralization reactions, while non-enzymatic antioxidants can come from natural materials such as fruits, onions and others. These natural materials contain several compounds that have antioxidant activity such as flavonoids, alkaloids, carotenoids and phenolic groups [111]. Testing of antioxidant activity can be carried out by some test methods such as DPPH free radical scavenging assay, oxygen radical absorbance capacity (ORAC) assay, trolox equivalent antioxidant capacity (TEAC) assay, ferric reducing antioxidant power (FRAP) assay, cupric reducing antioxidant capacity (CUPRAC) assay, reducing power assay and other methods [112,113,114].

Antioxidant compounds can also be obtained from some *Allium* species such as *A. fistulosum*, *A. ursinum*, *A. schoenoprasum*, *A. flavum*, *A. cepa*, *A. scorodoprasum*, *A. sativum*, *A. cepa* and *A. vineale* [69,70,102]. These compounds can be isolated from all parts of the plant such as bulbs, leaves, roots, flowers and bark [109]. This study will discuss the antioxidant activity of the compounds contained in *Allium* species leaves.

Testing of antioxidant activity in the species *A. sativum* was carried out using the DPPH and FRAP assay. Some studies reported that the antioxidant activity of the *A. sativum* leaves is very high, with IC_50_ 7.21 ± 0.39 mg/mL in the DPPH assay and 7.99 mol/g in the FRAP assay [114]. In 2005, Kim et al. succeeded in isolating four flavonol compounds from *A. sativum* leaves. These compounds are quercetin 3-*O*-β-D-glucopyranoside (isoquercitrin), quercetin 3-*O*-β-D-xylopyranoside (reynoutrin), kaempferol 3-*O*-β-D-glucopyranoside (astragalin) and isorhamnetin 3-*O*-β-D-glucopyranoside. The four compounds were tested for their antioxidant activity using the DPPH method, hydroxyl radical-scavenging activity and the ferric thiocyanate method [57]. Singh and Kumar also reported the presence of antioxidant activity in *A. sativum* leaves using the phosphomolybdenum reduction assay. The method is based on the reduction of Mo (IV) to Mo (V) in the methanol extract by formatting a green phosphate complex subsequence or Mo (V) [102].

El Hadidy et al. reported that there were three major compounds isolated from *A. fistulosum* leaf extract. They are myricetin, quercetin and rutin. From the three compounds, myricetin is the most abundant compound in the Giza 6 and photon varieties, among other compounds, which is 38.75%. The antioxidant activity test using the DPPH method showed that the activity decreased after three months of storage based on the percentage of antioxidants [61]. The ethanol extract of *A. ursinum* leaves, which also uses the DPPH radical scavenging assay, showed antioxidant activity of 77% with an EC_50_ value 322 g/mL. The activity was influenced by the presence of phenolic compounds in the extract [69]. Research on antioxidant activity was also carried out on *A. schoenoprasum* leaves which used two methods, DPPH bleaching assay and TEAC. The results using DPPH showed weak antioxidant activity with an EC_50_ value (6.72 ± 0.44 g/mg), whereas the TEAC method used to determine the total oxidant scavenging activity showed a value of 132.8 ± 23 g Trolox eq./g [64].

## 6. Structure-Antioxidant Activity Relationship Compounds in *Allium*

The structure–activity relationship (SAR) is an approach used to determine the relationship between the structure of a compound and its bioactivity [115]. The presence of certain substituents can affect the strength of compound activity; for example, the different number and position of a hydroxyl group will provide different antioxidant activities [116]. The following are some of the compound structures that have been isolated from *Allium* leaf extract: (Figure 10).

### 6.1. Apigenin

Apigenin (4′,5,7-trihydroxy-flavone) is a compound that has been isolated from the *A. fistulosum* leaves, such as in a study conducted by Immaculate V. et al. This compound shows a bright orange color when observed under UV light with an Rf value (0.83). Based on HPLC analysis data, this compound showed one major peak at 2.629 min and nine minor peaks (10.700; 13.604; 15.744; 17.324; 18.579; 19.525; 19.683; 21.360; 25.383), in minutes [34]. Li et al. reported that apigenin has low antioxidant activity, which is due to the absence of a single hydroxyl group in ring A and a single hydroxyl group in ring B with an activity value (1.5 mM) [122].

### 6.2. Myricetin

Myricetin (3′,4′,5′,3,5,7-hexahydroxyflavone) was also isolated from *A. fistulosum*. This compound is classified into the flavonoid group, which has six hydroxyl groups at positions 3, 5, 7, 3′, 4′ and 5′ [117]. The presence of a hydroxyl group at position 5′ in ring B greatly affects its antioxidant activity so that it becomes stronger with the IC_50_ (4 µM) and 463.40 ± 22.28 µM in testing using DPPH radical scavenging activity [118,120,123].

### 6.3. Naringenin

Naringenin 4′,5,7-trihydroxyflavanon is a flavanone compound of the flavonoid group with a molecular weight of 272.26 (C_15_H_12_O_5_) [124,125,126,127]. This compound had been isolated from *A. fistulosum* [61]. It has a saturated heterocyclic ring C with hydroxyl substituent at positions 4′,5,7. The presence of a hydroxyl group in ring A and a single hydroxyl group in ring B affect the naringenin value in the TEAC test (1.5 ± 0.05 mM), so that its antioxidant activity is lower than that of quercetin which has two hydroxyl substituents in ring B [122,124].

### 6.4. Kaempferol

Kaempferol can be found in fruits and vegetables [102,115]. This compound is also easily found in some *Allium* species [119]. In recent years, several studies had reported the presence of kaempferol in *A. fistulosum*, *A. ursinum*, *A. schoenoprasum*, *A. sativum* and other species [40,103]. Kaempferol (3,4′,5,7-tetrahydroxyflavone) is a yellow tetrahydroxyflavone compound that belongs to the flavonoid group with hydroxyl groups at positions 3, 4′, 5, and 7 [115,122,127]. Kaempferol has a wavelength band (367 nm) which is longer than compounds that only have three hydroxyl groups such as apigenin (337 nm). The presence of a reduction of the 2,3-unsaturated bond in ring C did not affect its antioxidant activity, whereas the presence of a single hydroxyl group in ring B, which is conjugated with a conjugated double bond, has little effect on increasing antioxidant activity [124]. Farkas et al. reported that kaempferol has antioxidant activity in inhibiting heat-induced oxidation in a β-carotene-linoleic acid-model-system (65.3%) [128]. In testing using the DPPH radical scavenging activity method, kaempferol has an IC_50_ value 28.05 ± 0.28 µM and 1.3 ± 0.08 mM in tests using the TEAC (Trolox equivalent antioxidant activity) [129,130].

### 6.5. Catechin

Catechin (3,3′,4′,5,7-pentahydroxyflavan) is a compound commonly found in several types of green tea, cocoa, red grapes and onions [49,120,131,132,133]. It also can be found and isolated from *A. schoenoprasum* [62]. This compound belongs to flavanol compound groups which have five hydroxyl substituents at positions 3, 3′, 4′, 5 and 7 [133]. In recent years, this compound had been reported to have antioxidant activity [125,133]. Silva et al. reported that catechin had antioxidant activity of 1.9 ± 0.1 µmol in the DPPH radical scavenging assay and 1.4 ± 0.3 µM trolox equivalents/µM flavonoids in the ORAC_ROO_ assay. *o*-catechol group in ring B showed a good effect on antioxidant activity [134]. In addition, the high planarity due to the intramolecular hydrogen bonding between the 3-OH and 6′-H substituents in flavanol compounds such as catechin can also provide a good antioxidant activity [135,136].

### 6.6. Quercetin

Quercetin (3,5,7,3′,4′-pentahydroxyflavon) is one of the flavonoid compounds found in plants such as onions, apples, berries and others. Its presence can be easily found in several *Allium* species such as *A. cepa*, *A. sativum*, *A. ursinum* and *A. fistulosum* [4,56]. This flavonol compound consists of three rings and five hydroxyl groups [123,137,138]. Several studies reported that quercetin has the ability as an antioxidant in reducing the formation of ROS [139,140,141,142,143,144,145]. The strength of antioxidant activity depends on the number of hydroxyl groups possessed, such as quercetin, which will provide stronger activity than naringenin and apigenin with three hydroxyl groups [116,117,121]. This is indicated by the value of IC_50_ that is smaller and TEAC values that are larger compared to the two compounds with each IC_50_ (10.89 ± 0.03; >1000; 463.40 ± 22.28 µM) and TEAC (4.7, 1.53, 1.45 mM) [118,119,120,121,122,123,124,125,126,127,128,129,130,131,132,133,134,135,136,137,138,139,140,141,142]. At the same time, the formation of a resonance-stabilized quinone structure due to the hydroxyl group adjacent to the ring C [125]. Antioxidant activity will decrease when there is glycolation of the hydroxyl group at position 3 on ring C [121].

## 7. Conclusions

*Allium* species such as *A. cepa*, *A. sativum*, *A. fistulosum*, *A. schoenoprasum*, *A. ursinum*, *A. flavum*, *A. scorodoprasum, A. vineale* and *A. atroviolaceum* have a great role in the health field. Those contain secondary metabolites that have several bioactivities such as antioxidant, antimicrobial, antibacterial, antifungal, anti-inflammatory and others. Their bioactivities are influenced by certain structure and functional groups.

## Figures and Tables

**Figure 1 molecules-26-07175-f001:**
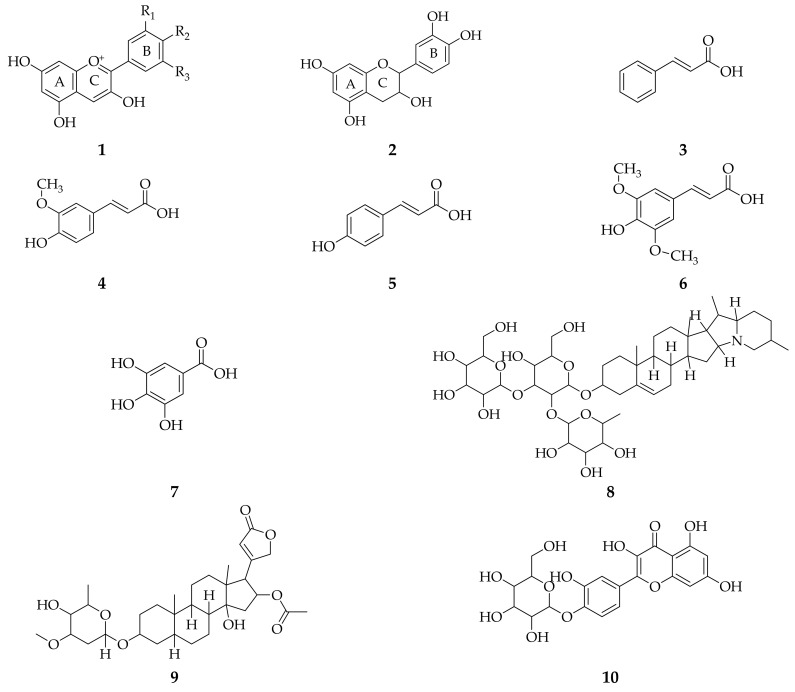
Compounds obtained from *A. cepa* leaf extract using different isolation methods. (**1**) anthocyanin [48]; (**2**) catechin; (**3**) cinnamic acid; (**4**) ferulic acid; (**5**) *p*-coumaric acid; (**6**) sinapic acid [40]; (**7**) tannin; (**8**) saponin [49]; (**9**) glycoside; (**10**) quercetin 4-O-β-glycopyranoside; (**11**) quercetin 3,4-O-β-diglycopyranoside; (**12**) quercetin 3,7,4-O-β-triglycopyranoside [7]; (**13**) β-caroten; (**14**) glutathione [27].

**Figure 2 molecules-26-07175-f002:**
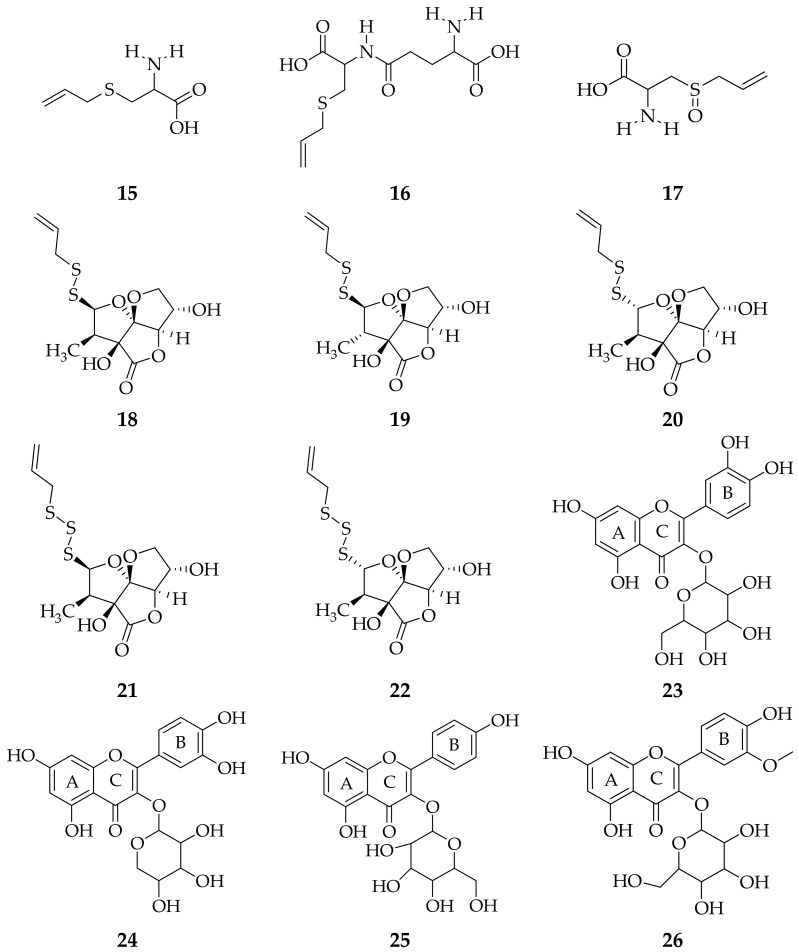
Compounds obtained from *A. sativum* leaf extract using different isolation methods. (**15**) *S*-allyl-L-cysteine; (**16**) γ-glutamyl-*S*-allyl-L-cysteine; (**17**) *S*-allyl-L-cysteine sulfoxide [55]; (**18**) foliogarlic disulfane A_1_; (**19**) foliogarlic disulfane A_2_; (**20**) foliogarlic disulfane A_3_; (**21**) foliogarlic trisulfane A_1_; (**22**) foliogarlic trisulfane A_2_ [52]; (**23**) quercetin 3-*O*-β-D-glucopyranoside; (**24**) quercetin 3-*O*-β-D-xylopyranoside; (**25**) kaempferol 3-*O*-β-D-glucopyranoside; (**26**) isorhamnetin 3-*O*-β-D-glucopyranoside [57].

**Figure 3 molecules-26-07175-f003:**
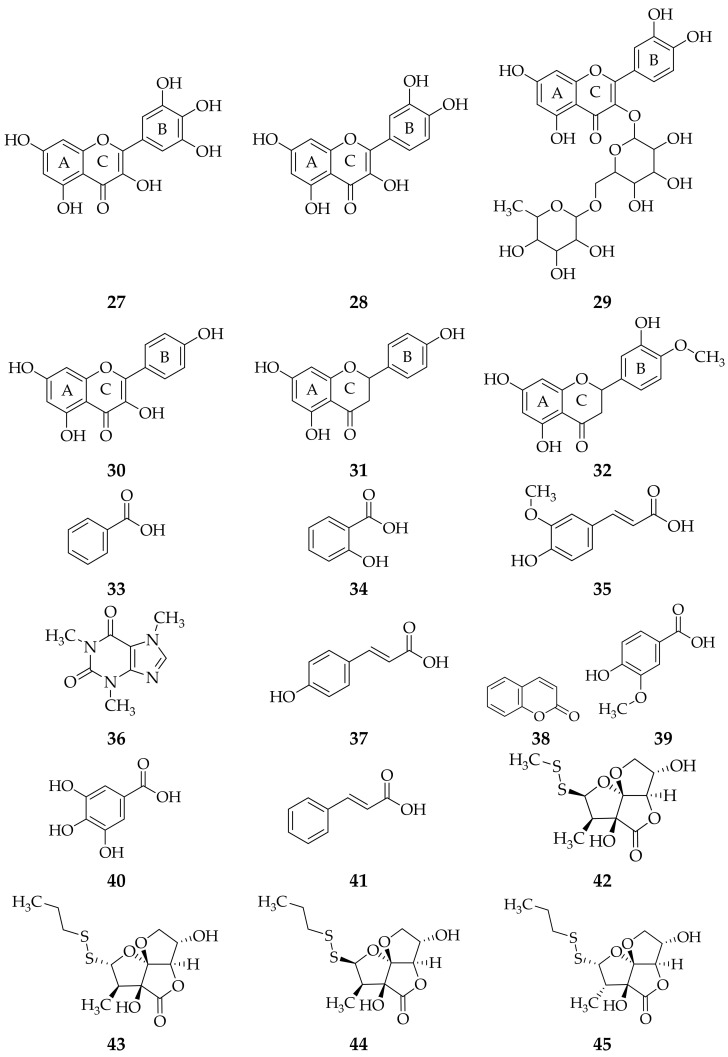
Compounds obtained from *A. fistulosum* leaf extract using different isolation methods. (**27**) myricetin; (**28**) quercetin; (**29**) rutin; (**30**) kaempferol; (**31**) naringenin; (**32**) hesperetin; (**33**) benzoic acid; (**34**) salicylic acid; (**35**) ferulic acid; (**36**) caffeine; (**37**) *p*-coumaric acid; (**38**) coumarin; (**39**) vanillic acid; (**40**) gallic acid; (**41**) cinnamic acid [61]; (**42**) kujounin A_3_; (**43**) kujounin B_1_; (**44**) kujounin B_2_; (**45**) kujounin B_3_; (**46**) allium sulfoxide A_2_; (**47**) allium sulfoxide A_3_; (**48**) kujounin A_1_ [59]; (**49**) apigenin [35]; (**50**) dichloroacetic acid; (**51**) 1-buten-3-yne; (**52**) α-pinene; (**53**) β-pinene; (**54**) D-limonene; (**55**) thymol [59]; (**56**) onionin A_1_; (**57**) onionin A_2_; (**58**) onionin A_3_ [60].

**Figure 4 molecules-26-07175-f004:**
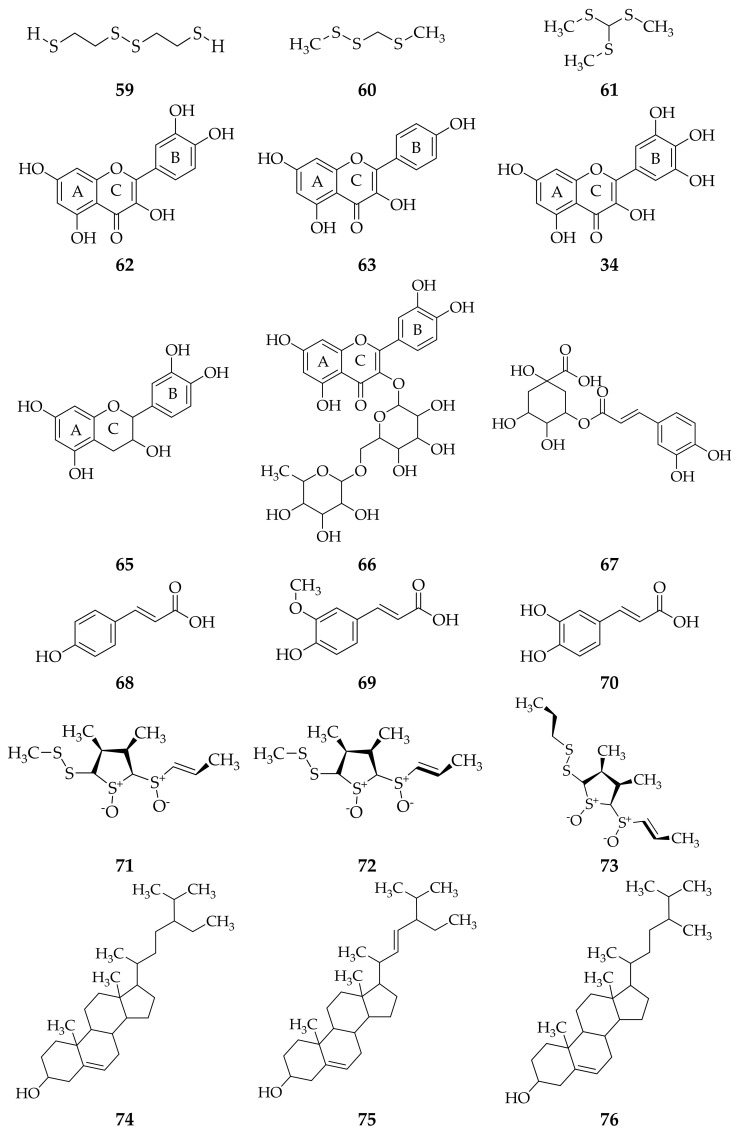
Compounds obtained from *A. schoenoprasum* leaf extract using different isolation methods. (**59**) *bis*-(2-sulfhydryethyl)-disulfide; (**60**) 2,4,5-trithiahexane; (**61**) tris(methylthio)-methane; (**62**) quercetin; (**63**) kaempferol; (**64**) myricetin; (**65**) catechin; (**66**) rutin; (**67**) chlorogenic acid; **(68)**
*p*-coumaric acid; (**69**) ferulic acid; (**70**) caffeic acid [61,63]; (**71**) folionin A_1_; (**72**) folionin A_2_; (**73**) folionin B [66]; (**74**) sitosterol; (**75**) stigmasterol; (**76**) campesterol; (**77**) cholesterol; (**78**) free fatty acid; (**79**) monoacylglycerin; (**80**) diacylglycerin; (**81**) triacylglycerin; (**82**) linoleic acid; (**83**) palmitic acid; (**84**) spirostanols; (**85**) deltonin; (**86**) saponin; (**87**) furostanols [67]; (**88**) I-ketose [68]; (**89**) protodioscin [67].

**Figure 5 molecules-26-07175-f005:**
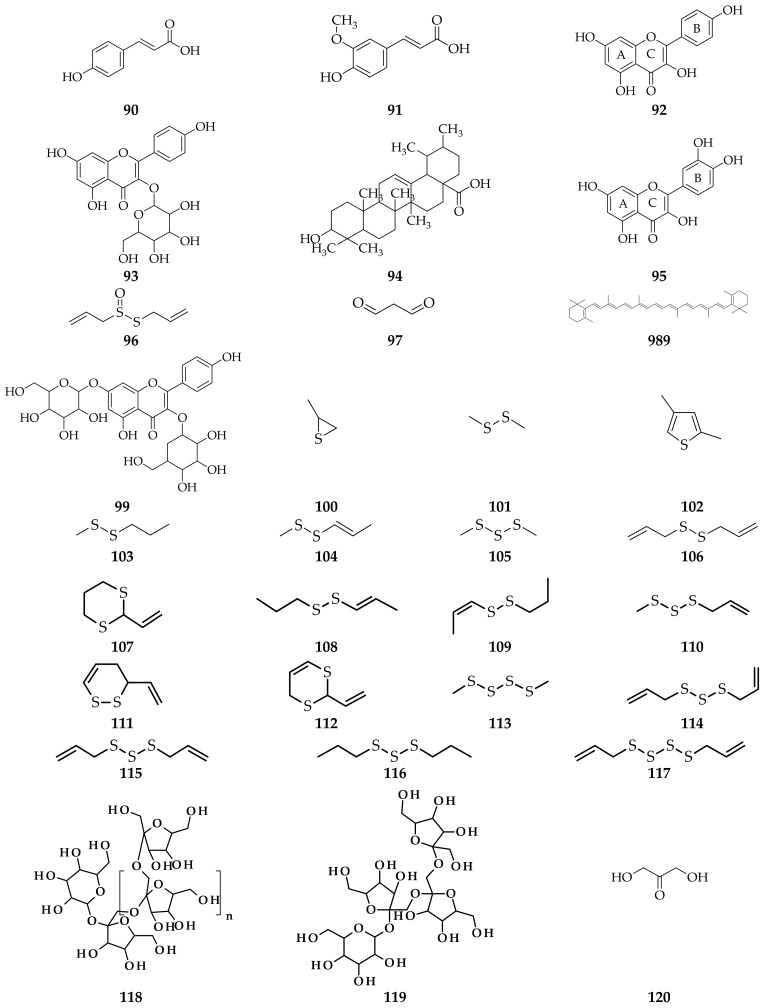
Compounds obtained from *A. ursinum* leaf extract using different isolation methods. (**90**) *p*-coumaric acid; (**91**) ferulic acid; (**92**) kaempferol; (**93**) kaempferol-3-*O*-glycoside; (**94**) ursolic acid; (**95**) quercetin [39]; (**96**) allicin [21]; (**97**) malondialdehyde; (**98**) β-caroten; (**99**) kaempferol-3,7-di-*O*-β-D-glucopyranoside [71]; (**100**) propylene sulfide; (**101**) dimethyldisulfide; (**102**) 2,4-dimethylthiophene; (**103**) (*E*)-methyl-2-propyl disulfide; (**104**) (*Z*)-methyl-2-propenyl disulfide; (**105**) dimethyl trisulfide; (**106**) di-2-propenyl disulfide; (**107**) 2-vinyl-1,3-dithiane; (**108**) (*E*)-propenyl propyl disulfide; (**109**) (*Z*)-propenyl propyl disulfide; (**110**) methyl-2-propenyl trisulfide; (**111**) 3,4-dihydro-3-vinyl-1,2-dithiine; (**112**) 2-vinyl-4H-1,3-dithiine; (**113**) dimethyl tetrasulfide; (**114**) (*E*)-di-2-propenyl trisulfide; (**115**) (*Z*)-di-2-propenyl trisulfide; (**116**) di-2-propyl trisulfide; (**117**) di-2-propenyl tetrasulfide [72]; (**118**) inulin; (**119**) nystose; (**120**) I-ketose [68].

**Figure 6 molecules-26-07175-f006:**
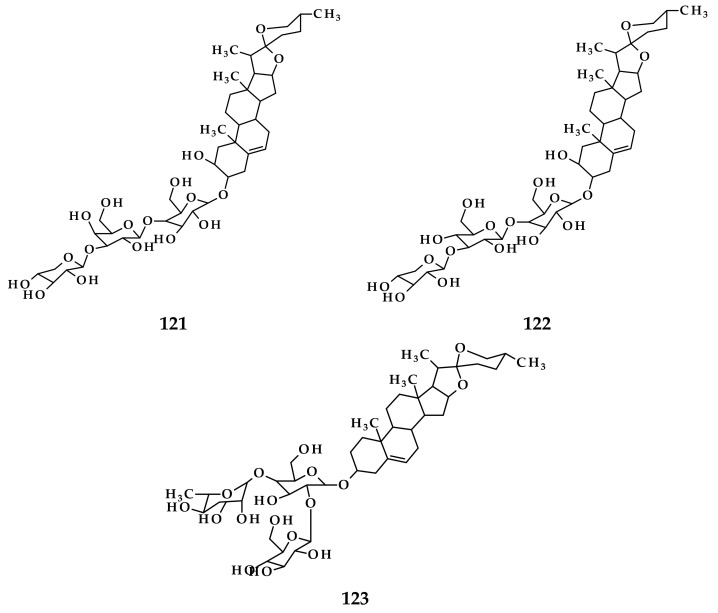
Three new compounds of spirostane-type glycosides obtained from *A. flavum* leaf extract: (**121**) (20*S*,25*R*)-2α-hydroxyspirost-5-en-3β-yl; (**122**) (20*S*,25*R*)-2α-hydroxyspirost-5-en-3β-yl; (**123**) (20*S*,25*R*)-spirost-5-en-3β-yl [11].

**Figure 7 molecules-26-07175-f007:**
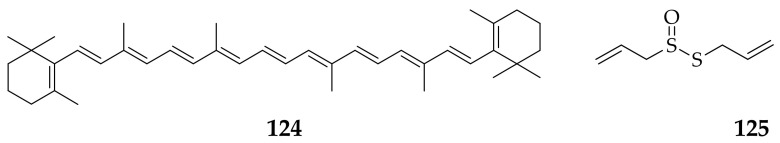
Compounds obtained from *A. scorodoprasum* leaf extract using ultrasonic bath: (**124**) β-caroten [27]; (**125**) allicin [12].

**Figure 8 molecules-26-07175-f008:**
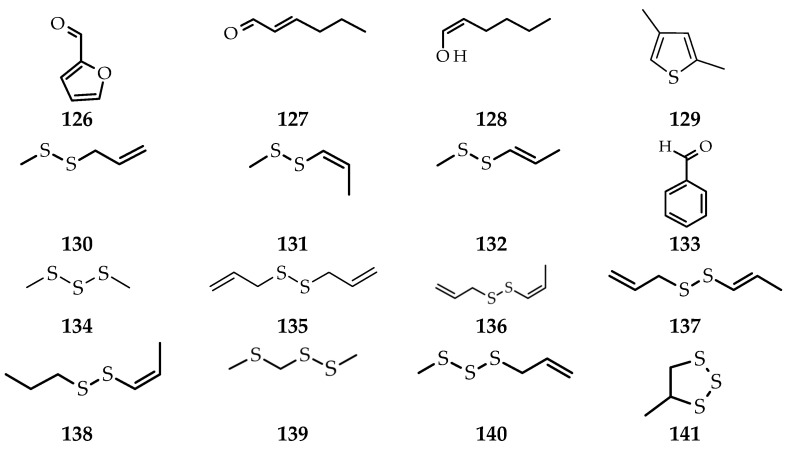
Compounds obtained from *A. vineale* leaf extract using different isolation methods. (**126**) 2-furaldehyde; (**127**) (2*E*)-hexenal; (**128**) (3*Z*)-hexenol; (**129**) 2,4-dimethylthiophene; (**130)** allyl methyl disulfide; (**131**) methyl (*Z*)-1-propenyl disulfide; (**132**) methyl (*E*)-1-propenyl disulfide; (**133**) benzaldehyde; (**134**) dimethyl trisulfide; (**135**) diallyl disulfide; (**136**) allyl (*Z*)-1-propenyl disulfide; (**137**) allyl (*E*)-1-propenyl disulfide; (**138**) 1-propenyl propyl disulfide; (**139**) methyl (methylthio)methyl disulfide; (**140**) allyl methyl trisulfide; (**141**) 4-methyl-1,2,3-trithiolane; (**142**) methyl propyl trisulfide; (**143**) methyl (*Z*)-1-propenyl trisulfide; (**144**) methyl (*E*)-1-propenyl trisulfide; (**145**) dimethyl tetrasulfide; (**146**) allyl (methylthio)methyl disulfide; **(147)** diallyl trisulfide; (**148**) allyl (*Z*)-1-propenyl trisulfide; (**149**) *p*-vinylguaiacol; (**150**) allyl propyl trisulfide; (**151**) 5-methyl-1,2,3,4-tetrathiane; (**152**) methyl (methylthio)methyl trisulfide; **(153)** allyl methyl tetrasulfide; (**154**) allyl (methylthio)methyl trisulfide; (**155**) 4-methyl-1,2,3,5,6-pentathiepane [45]; (**156**) chrysoeriol; (**157**) chrysoeriol-7-O[200-*O*-*E*-feruloyl]-β-D-glucoside; (**158**) isorhamnetin-3-*O*-β-D-glucoside [16].

**Figure 9 molecules-26-07175-f009:**
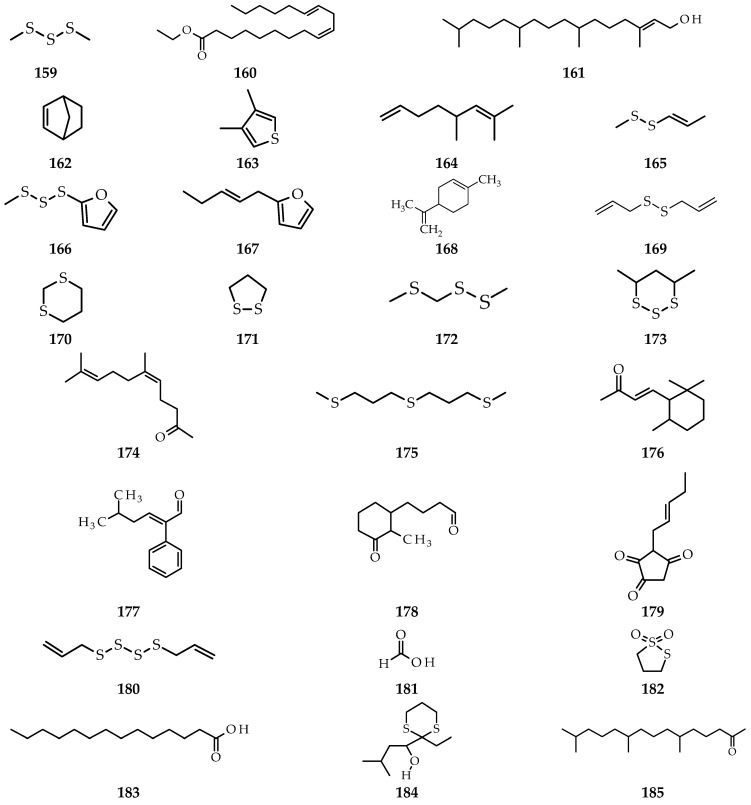
Compounds obtained from *A. atroviolaceum* leaf extract using different isolation methods. (**159**) dimethyl trisulfide; (**160**) ethyl linolenate; (**161**) phytol [73]; (**162**) norbornene; (**163**) 3,4-dimethylthiophene; (**164**) isocitronellene; (**165**) methyl 1-propenyl disulfide; (**166**) dimethyl trisulfide furan; (**167**) *trans*-2-(2-pentenyl)-furan; (**168**) limonene; (**169**) diallyl disulfide; (**170**) 1,3-dithiane; (**171**) 1,2-dithiolane; (**172**) methyl (methylthio)methyl disulfide; (**173**) 4,6-dimethyl-[1–3]-trithiane; (**174**) 5,9-undecadien-2-one, 6,10-dimethyl-(*Z*); (**175**) 1,1′-thiobis-3-(methylthio)-propane; (**176**) 4-(2,6,6-trimethyl-1-cyclohexen-1-yl)-3-buten-2-one; (**177**) 5-methyl-2-phenyl-2-hexenal; (**178**) 2-methyl-3-oxo-*cis*-cyclohexanebutanal; (**179**) 1,2,4-cyclopentanetrione,3-(2-pentenyl)-; (**180**) di-2-propenyl tetrasulfide; (**181**) formic acid; (**182**) 1,2-dithiolane,1.1-dioxide; (**183**) tetradecanoic acid; (**184**) 1-(2-ethyl-[1,3]dithian-2-yl)-3-methyl-butan-1-ol; (**185**) 6,10,14-trimethylpentadecan-2-one [74].

**Figure 10 molecules-26-07175-f010:**
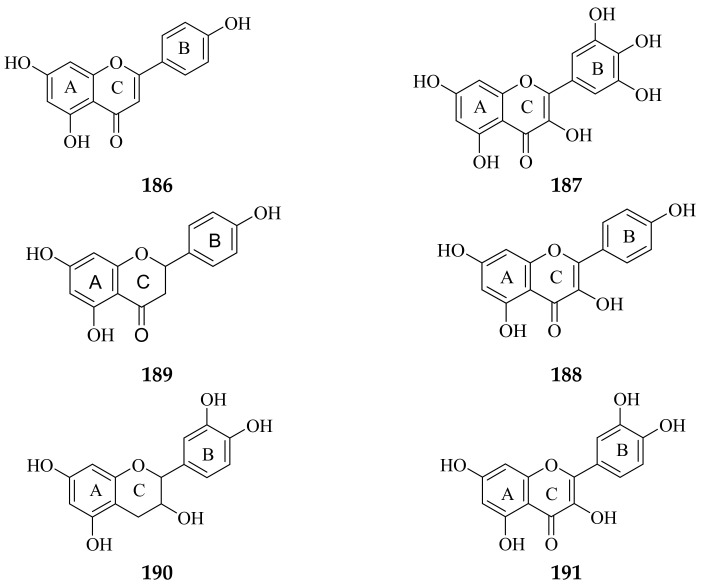
Compound structures are successfully isolated from *Allium* leaf extracts. (**186**) apigenin [35]; (**187**) myricetin [117]; (**188**) naringenin [118]; (**189**) kaempferol [119]; (**190**) catechin [120]; (**191**) quercetin [121].

## Data Availability

The study did not report any data.

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
