# Peer review of "Antioxidant Properties and Structure-Antioxidant Activity Relationship of Allium Species Leaves"

_molecules, 2021, doi:10.3390/molecules26237175_

Round 1

Reviewer 1 Report

The manuscript entitled “Antioxidant Properties and Structure-Antioxidant Activity Re-2 lationship of Allium Species Leave” by Kurnia et al., reviewed the bioactive compounds isolated from Allium species. Authors reviewed the detailed information on the structure and bioactivities of compounds from Allium. While this review covered the general interests on the useful metabolites from important Allium species, I think that this manuscript can be considered to be published in the ‘Molecules’ with proper response to the raised issues as below together with English editing.

Comments

  1. Line 17

secondary metabolites compounds

--> secondary metabolites (and throughout the manuscript)

  1. Line 18

as antioxidants, antimicrobials, antibacterial, antifungal,

--> as antioxidants,  antibacterial, antifungal, (and throughout the manuscript)

  1. Line 19

pancreatic α-amylase inhibitors and glucoamylase enzymes

--> pancreatic α-amylase inhibitors and glucoamylase enzymes

--> Dose ‘glucoamylase enzymes’ mean glucoamylase activity or inhibitor? Please clarify this.

  1. Lines 35-36

Several studies reported that the leaves of the Allium species have a good antioxidant activity.

--> Please add proper references.

  1. Lines 36-37

Considering that Allium leaf is a natural material sometimes considered as waste and has not been utilized optimally.

--> This sentence is not adequate for the paragraph and better to be removed.

  1. Line 64

the plant is also reported to have heart disease

--> the plant is also reported to have heart disease preventing effect.

  1. Line 66

Allium species known originating from

--> Allium species known for originating from

  1. Line 72

Is it proper to mention ‘Canada’ considering the previous sentence referring Central Europe?

  1. Line 79

in East Asia used to relieve flu and lung congestion

--> in East Asia, this is used to relieve flu and lung congestion

  1. Line 80

ramsson

--> ramsons

  1. Line 82

Bulbs measuring less than 6 cm

--> bulbs with the size of less than 6 cm

  1. Line 83

dyspepsia disorders

--> dyspepsia

  1. Line 121

samples were

--> Samples were

  1. Line 125

Of the some different methods, A. cepa leave contains

--> From some different isolation methods, it was also reported that A. cepa leave contains

  1. Figure 2 and 3

--> I suggest to combine these two figures as one.

  1. Figure 5~7

--> I suggest to combine these three figures as one.

  1. Figure 8~11

--> I suggest to combine these four figures as one.

  1. Line 249

from the some studies with

--> from some studies with

  1. Line 292

Based on study

--> Based on the study

  1. Line 296

was determined by

--> was conducted by

  1. Figure 12~13

--> I suggest to combine these two figures as one. 

  1. Line 344

Rezgui et al. (2014) also conducted

--> Please verify the correct Reference formatting.

  1. Line 373

Several studies reported that isolation of A. vineale used some methods.

--> Several methods were used to isolate compounds from A. vineale. (throughout the manuscript in the similar cases)

  1. Figure 16~17

--> I suggest to combine these two figures as one.

  1. Figure 18~19

--> I suggest to combine these two figures as one.

  1. Line 483

MIC

--> mini- 491 mum inhibitory concentrations (MIC) (Please define MIC at its first use in the manuscript and then use abbreviation from next, for example in line 491, as MIC)

  1. Line 494

Other studies also state that in A. ursinum leaf extracts contain

--> Other studies also state that A. ursinum leaf extracts contain

  1. Lines 511-512

The unit of l/ml seems to be incorrect. Does it mean ul/ml? Please verify this.

  1. Line 516

reported his research on antioxidant activity tests on A. schoenoprasum that used the

--> reported the antioxidant activity of A. schoenoprasum based on the

  1. Lines 547-548

the enzymes CAT, SOD,  GPX

--> Please inform the full name of each enzymes for their first use in the manuscript.

  1. Lines 605-606

stress caused by free radical reactions and

--> stress caused by free radical and

  1. Lines 609-610

While free radicals are atoms, molecules or unpaired electrons that can easily react with other molecules.

--> This sentence can be removed.

  1. Figure 20, equation 1 and2

These information is very general one and if authors can prepare more detailed and specified for Allium species, it will be much better. Otherwise, these can be omitted.

  1. Figure 21~22

--> I suggest to combine these two figures as one.

  1. Line 686
  2. fistulosum leave conducted by V. et al

--> in complete author or reference information. Please clarify this.

  1. Line 687

one major peak at 2,629 minutes

--> one major peak at 2.629 minutes ?

  1. Hydroxyl  group or OH group, please unify this throughout the manuscript.
  2. Section 6.5

Can authors give the examples of the isolation or presence of catechin from Allium species in this paragraph?

  1. Section 6.6

Can authors give the examples of the isolation or presence of quercetin from Allium species in this paragraph?

  1. Section 7

I suggest to rewrite the conclusion to summarize the importance of the compounds isolated from Allium species in more clear and concise way.

Author Response

Dear,

We are grateful to reviewers and editors for their time and constructive comments on our manuscript.

We have implemented their comments and suggestion and wish to submit a revised version of the manuscript for further consideration in the journal.

We look forward to the outcome of your assessment and we hope that our manuscript in this journal could be accepted.

Sincerely yours,

Dikdik Kurnia

Reviewer 2 Report

The authors have presented a very thorough review of the properties of the compounds originating from the leaves of  Allium species.  This review summarizes nine types of Allium species (ethnobotany and  ethnopharmacology),  the content of compounds of Allium species leaves with various isolation  methods, bioactivities, antioxidant properties, and structure-antioxidant activity relationship of Allium compounds. Based on the summarized information, the authors concluded that different types of Allium species such as A. cepa, A. sativum, A. fistulosum, A. schoenoprasum, A. ursinum, A. flavum, A. scorodoprasum, A. vineale, A. atroviolaceum have great benefits in the health field. All parts of the plants have different bioactivities (as antioxidant, antimicrobial, antibacterial, antifungal, anti-inflammatory, and others) and are widely used in traditional medicine. 

The manuscript is well written and the data are well discussed and easy to follow by experts and non-experts in the field. I recommend it for publication without any modifications.

Author Response

Dear,

Thank you and appreciations for your great corrections, suggestions and recommendations

Sincerely your,

Dikdik Kurnia
